# Inodilators May Improve the In-Hospital Mortality of Patients with Cardiogenic Shock Undergoing Veno-Arterial Extracorporeal Membrane Oxygenation

**DOI:** 10.3390/jcm11174958

**Published:** 2022-08-24

**Authors:** Mijoo Kim, Seok-Woo Seong, Pil Sang Song, Jin-Ok Jeong, Jeong Hoon Yang, Hyeon-Cheol Gwon, Young-Guk Ko, Cheol Woong Yu, Woo Jung Chun, Woo Jin Jang, Hyun-Joong Kim, Jang-Whan Bae, Sung Uk Kwon, Hyun-Jong Lee, Wang Soo Lee, Sang-Don Park, Sung Soo Cho, Jae-Hyeong Park

**Affiliations:** 1Division of Cardiology, Department of Internal Medicine, Chungnam National University Hospital, Daejeon 35015, Korea; 2Division of Cardiology, Department of Medicine, Heart Vascular Stroke Institute, Samsung Medical Center, Sungkyunkwan University School of Medicine, Seoul 06351, Korea; 3Division of Cardiology, Severance Cardiovascular Hospital, Yonsei University College of Medicine, Seoul 03722, Korea; 4Division of Cardiology, Department of Internal Medicine, Korea University Anam Hospital, Seoul 02841, Korea; 5Department of Cardiology, Samsung Changwon Hospital, Sungkyunkwan University School of Medicine, Changwon 51353, Korea; 6Department of Cardiology, Ewha Woman’s University Seoul Hospital, Ehwa Woman’s University School of Medicine, Seoul 07804, Korea; 7Division of Cardiology, Department of Medicine, Konkuk University Medical Center, Seoul 05030, Korea; 8Department of Internal Medicine, Chungbuk National University College of Medicine, Cheongju 28644, Korea; 9Division of Cardiology, Department of Internal Medicine, Ilsan Paik Hospital, University of Inje College of Medicine, Seoul 48108, Korea; 10Division of Cardiology, Department of Medicine, Sejong General Hospital, Bucheon 14754, Korea; 11Division of Cardiology, Department of Medicine, Chung-Ang University Hospital, Seoul 06973, Korea; 12Division of Cardiology, Department of Medicine, Inha University Hospital, Incheon 22332, Korea; 13Division of Cardiovascular Medicine, Department of Internal Medicine, Dankook University Hospital, Dankook University College of Medicine, Cheonan 31116, Korea

**Keywords:** extracorporeal membrane oxygenation, cardiogenic shock, dobutamine, milrinone

## Abstract

Although inodilators (dobutamine and milrinone) are widely used empirically for cardiogenic shock (CS), the efficacy of inodilators for patients with CS undergoing veno-arterial extracorporeal membrane oxygenation (VA-ECMO) is controversial. We evaluated the effects of inodilators on clinical outcomes using the RESCUE (REtrospective and prospective observational Study to investigate Clinical oUtcomes and Efficacy of left ventricular assist device for Korean patients with cardiogenic shock; NCT02985008) registry. We selected and analyzed the clinical outcomes of 496 patients who underwent VA-ECMO and did or did not receive inodilators. Of the 496 patients, 257 (51.8%) died during hospitalization. We selected 191 matched pairs to adjust for baseline clinical characteristics after 1:1 propensity score matching (PSM). The univariate and multivariate analyses showed that the inodilator group had significantly lower in-hospital mortality than the no-inodilator group (unadjusted hazard ratio [HR], 0.768; 95% confidence interval [CI], 0.579–1.018; *p* = 0.066, adjusted HR, 0.702; 95% CI, 0.552–0.944; *p =* 0.019). For patients with CS undergoing VA-ECMO, inodilators may improve clinical outcomes.

## 1. Introduction

Cardiogenic shock (CS) is circulatory failure related to systemic hypoperfusion and end-organ dysfunction resulting from a low cardiac output state. Early correction of hemodynamic instability may prevent the evolution of multiple organ failure and reduce mortality [1]. Although advances in pharmacological and mechanical circulatory support have reduced the in-hospital mortality of patients with CS, the mortality rate remains high [2,3]. Specifically, refractory CS, which is persistent CS despite volume resuscitation or the use of vasopressors or inotropes, has a case fatality rate of more than 50% [4].

Veno-arterial-extracorporeal membrane oxygenation (VA-ECMO) is a therapeutic option for stabilizing patients with refractory CS. Furthermore, VA-ECMO is an alternative to conventional medical therapy and is expected to improve patient survival. The number of patients treated with VA-ECMO has increased exponentially over the past decade [5], and several studies have shown its beneficial effects on the prognosis of patients with CS [6,7].

Currently, inotropes or vasopressors are used empirically for many patients with refractory CS undergoing VA-ECMO. Dobutamine and milrinone are among the most common inodilators used for patients with CS. Inodilators increase the stroke volume and cardiac output by directly stimulating myocardial contractility and reducing afterload through systemic vasodilation. Using inodilators in conjunction with VA-ECMO can increase left ventricular ejection, thereby reducing the risk of intracardiac stasis. However, the clinical effects of inodilators on these patients are controversial. This is mainly because of the lack of alternative therapies and prospective randomized controlled trials. There is uncertainty about whether inodilators may benefit patients with CS treated with VA-ECMO. Therefore, we analyzed the in-hospital outcomes of patients with refractory CS treated using VA-ECMO to determine the beneficial effects of inodilators (dobutamine and milrinone).

## 2. Materials and Methods

### 2.1. Study Population

The RESCUE (REtrospective and prospective observational Study to investigate Clinical oUtcomes and Efficacy of left ventricular assist device for Korean patients with cardiogenic shock; NCT02985008 at www.clinicaltrials.gov, assessed on 5 December 2016) registry is a multicenter registry. This study was conducted retrospectively and prospectively and included patients with CS aged > 19 years old from January 2014 to December 2018. In this registry, 1247 patients were enrolled from 12 tertiary centers in the Republic of Korea (954 retrospectively and 293 prospectively). According to the IABP-shock II trial definition, we included patients who had a systolic blood pressure (SBP) < 90 mmHg for 30 min or needed inotrope or vasopressor support to achieve an SBP > 90 mmHg and the presence of pulmonary congestion and signs of impaired organ perfusion (altered mental status, cold skin and extremities, urine output <0.5 mL/kg/h for the previous 6 h, or serum lactate > 2 mmol/L). The major exclusion criteria were out-of-hospital cardiac arrest, shock caused by factors other than CS (hypovolemic shock, septic shock, or post-cardiotomy shock), and refusal of active treatment.

### 2.2. Data Collection and Outcomes

The Institutional Review Board (IRB) of each hospital approved the study protocol. This study was conducted in accordance with the latest principles of the Declaration of Helsinki. The IRBs of the participating hospitals waived the requirement for informed consent from the retrospectively enrolled patients. However, we obtained written informed consent from all prospectively enrolled patients before enrollment. Independent clinical research coordinators collected patient data from web-based case report forms. Additional clinical information was obtained from medical records if necessary. The primary outcome was the in-hospital mortality rate. The secondary outcomes included in-hospital cardiac mortality and complications related to VA-ECMO, such as bleeding, limb ischemia, cerebrovascular accidents, and sepsis. All-cause mortality was defined as death from any cause, and all deaths were considered attributable to cardiac causes unless a non-cardiac cause was certain. The vasoactive-inotropic score was calculated as the maximal administration rate of dopamine, dobutamine, epinephrine, norepinephrine, milrinone, and vasopressin during the 48-h period of the shock event.

### 2.3. ECMO Management

VA-ECMO was initiated for patients with prolonged and recurrent arrest or with severe therapy-resistant CS [8]. The VA-ECMO device was inserted by percutaneous cannulation using Seldinger’s technique or surgical cannulation using the cut-down method at the femoral vessels. For patients without life-threatening bleeding, anticoagulation was provided intravenously using unfractionated heparin with an activated clotting time of 180 to 200 s during ECMO support. Management of the volume and vasopressors or inotropes was based on the clinical judgment of the attending physician in charge. Additionally, the decision to administer inodilators (dobutamine and milrinone) was based on the clinical judgment of the attending physician.

### 2.4. Statistical Analysis

Continuous variables are presented as the mean ± standard deviation and were compared using Student’s *t*-test. Categorical variables are presented as numbers and percentages and were compared using the χ^2^ test. We calculated the propensity score to match the baseline characteristics between the groups with and without inodilators use. Patients were matched 1:1. Univariate and multivariate time-dependent Cox-proportional hazard analyses were performed to predict the in-hospital mortality of the entire cohort and the propensity score-matched cohort. Covariates that were statistically significant in the univariate analysis (*p* < 0.1) or considered clinically important were included in the multivariate models. Hazard ratios (HRs) were calculated as an estimate of the risk associated with a particular variable based on binomial distributions and presented with 95% confidence intervals (CIs). All probability values were two-tailed, and *p* < 0.05 was considered statistically significant. Statistical analyses were performed using SPSS software version 23 (IBM, Armonk, NY, USA).

## 3. Results

### 3.1. Baseline Characteristics According to the Use of Inodilators

This cohort included 1247 patients with CS, with 496 of them treated with VA-ECMO during the course of the study (Figure 1).

The mean age of the study population was 61.8 ± 14.2 years, and 69.2% were men. Of the 496 patients, 257 (51.8%) died during hospitalization, and 270 (54.4%) were treated with inodilators. The baseline characteristics of the total cohort and propensity-matched cohort are shown in Table 1.

Compared to the no-inodilator group, the inodilator group had a significantly lower left ventricular ejection fraction (26.2 ± 12.4% versus 29.0 ± 15.0%, *p* = 0.034) and a higher inotropic score (34.1 ± 41.5 versus 17.6 ± 40.9, *p* < 0.001). The total bilirubin levels were higher in the inodilator group than in the no-inodilator group (1.4 ± 3.3 versus 0.8 ± 0.9, *p* = 0.010). There were no other significant differences in baseline characteristics according to the use of inodilators, except for the low prevalence of diabetes mellitus in the inodilator group (83 (30.7%) versus 96 (42.5%), *p* = 0.007).

### 3.2. In-Hospital Outcomes of the Propensity Score Matched Analysis

After propensity score matching, 191 matched pairs (191 patients from the inodilator group and 191 patients from the no-inodilator group) were obtained. There were no significant differences in the baseline clinical characteristics, except for previous cerebrovascular accidents (Table 1). In the matched cohort, 194 (50.8%) patients died during hospitalization. The clinical outcomes are summarized in Table 2. There were no significant differences in in-hospital mortality, in-hospital cardiac mortality, ECMO cannula insertion site bleeding, limb ischemia, gastrointestinal bleeding, cerebrovascular accidents, or sepsis.

The results of the univariate and multivariate analyses of the in-hospital mortality in propensity score matched cohort are listed in Table 3. The univariate analysis revealed that age, heart rate, hypertension, previous chronic kidney disease, ischemic cardiomyopathy, creatinine clearance rate, lactic acid level, peak creatine kinase-myocardial band level, left ventricular ejection fraction, cardiopulmonary resuscitation, shock to ECMO insertion time, continuous renal replacement therapy, mechanical ventilation, ECMO initial pump flow, distal perfusion, and vasoactive inotropic score were significant determinants of in-hospital mortality. In the multivariate analysis, the use of inodilators was significantly associated with lower in-hospital mortality after adjustment (adjusted HR = 0.702, 95% CI = 0.552–0.944, *p* = 0.019) (Figure 2). Furthermore, age, lactic acid level, left ventricular ejection fraction, continuous renal replacement therapy, mechanical ventilation, distal perfusion, and vasoactive inotropic score remained statistically significant after adjustment.

## 4. Discussion

We showed that the use of inodilators was beneficial for patients with CS treated with VA-ECMO.

Inotropic agents have been fundamental for resuscitation cases of CS for several decades. In patients with CS, the blood pressure may fail to increase after the initial resuscitation, prompting the use of inotropic agents, as proposed by the current international guidelines [9,10]. Inotropic agents can be classified as inopressors and inodilators based on their effects on systemic vascular resistance. Dobutamine and milrinone are two inodilators approved for use in the United States [11]. Dobutamine is a synthetic catecholamine that has a direct agonistic effect on β_1_- and β_2_-adrenergic receptors, resulting in improved myocardial inotropy, weak chronotropy, and systemic vasodilation. Milrinone is a phosphodiesterase III inhibitor that increases intracellular cyclic adenosine monophosphate levels, resulting in improved cardiac contractility as well as systemic and pulmonary circulatory vasodilation. Therefore, both dobutamine and milrinone lead to increased cardiac output and decreased left ventricular filling pressure [12,13]. Previous studies have shown changes in hemodynamic parameters with increased cardiac index and decreased pulmonary capillary wedge pressure when inodilators were used [14,15,16]. 

VA-ECMO leads to an increase in the afterload [17,18]. As a result, it can increase left ventricular and atrial filling pressures, pulmonary edema, left ventricular distension, and stasis in the pulmonary circulation [19,20]. Inodilators can increase left ventricular performance; therefore, they can be used to treat stasis induced by VA-ECMO and reduce systemic and pulmonary vascular resistance. Therefore, the use of inodilators can be beneficial for patients undergoing VA-ECMO.

Pirracchio et al. reported that a combination of inodilators and inopressors was associated with a significant improvement in the short-term survival of patients with severe CS without ECMO compared with the use of inopressors alone [21]. These findings suggest that either excessive vasoconstriction can be detrimental to patients with severe CS or vasodilation may be beneficial.

Zotzmann et al. reported no statistically significant difference in the short-term survival of patients who received inodilators and those who did not receive inotropes. However, the survival rate of the group that received inodilators was 52.2%, which was higher than that of the group that did not receive inotropes (47.9%) [22].

Previous studies have compared the effects of dobutamine and milrinone. Mathew et al. reported that no significant differences were observed between milrinone and dobutamine in terms of the clinical outcomes, including in-hospital death, of patients with CS [23]. Lewis et al. also indicated that there were no significant differences in the efficacy and safety of milrinone and dobutamine [24]. Therefore, the efficacy and safety of both drugs are similar, and milrinone or dobutamine can be used as inodilators for CS, depending on the condition of the patient.

There are no convincing data supporting the use of inodilators to reduce the mortality of patients with refractory CS undergoing VA-ECMO. However, by using a large-scale multicenter registry, this study provided evidence that inodilators may be beneficial for these patients.

This study had several limitations. First, the RESCUE registry contains observational data; therefore, residual confounding variables may have affected our findings, even after multiple adjustments. Second, the in-hospital treatment reflects the preferences and discretion of individual physicians, which may have introduced confounding selection bias. Third, the corresponding action and adverse effects may differ depending on the dose, duration of use, and method of administration of dobutamine and milrinone [11,25]. However, this has not been considered in detail, and the adverse effects have not been properly evaluated. Additionally, hemodynamic parameters, such as cardiac output, cardiac index, central venous pressure, and pulmonary capillary wedge pressure, were not included. If hemodynamic changes attributable to the use of inodilators had been included in the data, concrete evidence for the clinical effect could have been presented. Furthermore, the study was restricted to in-hospital outcomes only; however, mid-term as well as long-term outcome assessments are required to confirm the results. Finally, fewer patients received milrinone and more patients received dobutamine; therefore, caution is advised when interpreting these results.

## 5. Conclusions

In-hospital mortality remains high for patients undergoing VA-ECMO for refractory CS. Although the use of inodilators should be considered based on clinical judgment, their use may improve the clinical outcomes of these patients. To confirm the favorable effects of inodilators on patients with refractory CS treated with VA-ECMO, future studies involving well-designed randomized controlled trials are required.

## Figures and Tables

**Figure 1 jcm-11-04958-f001:**
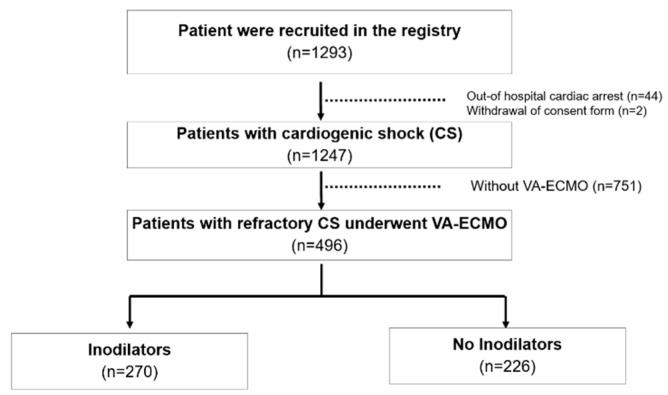
Study flow.

**Figure 2 jcm-11-04958-f002:**
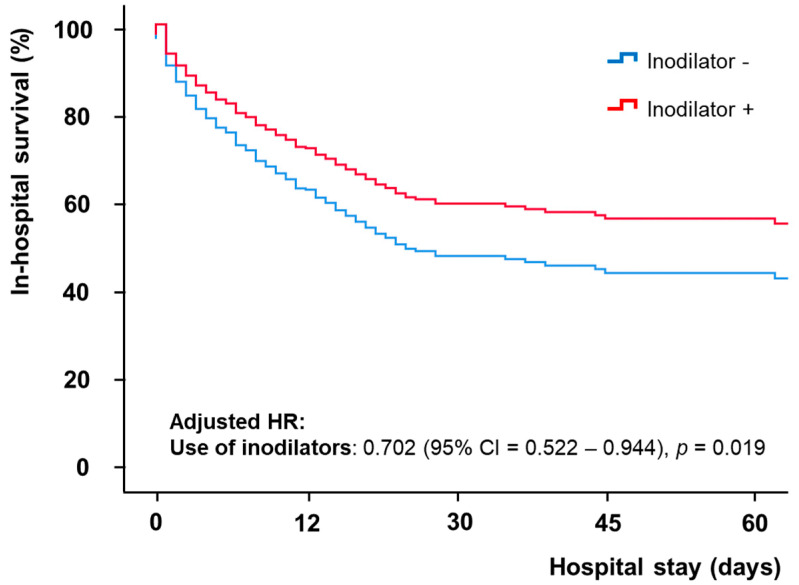
Comparison of in-hospital mortality according to the use of inodilators. The adjusted hazard ratio (HR) was calculated using the Cox-proportional hazard analysis to rule out the effects of other significant variables in the univariate analysis.

**Table 1 jcm-11-04958-t001:** Comparison of clinical characteristics according to the use of inodilators.

	Total Cohort (n = 596)			Propensity Score Matched Cohort (n = 382)		
	Inodilators (n = 270)	No Inodilators (n = 226)	*p* Value	Inodilators (n = 191)	No Inodilators (n = 191)	*p* Value
Age (year)	60.9 ± 14.8	62.9 ± 13.7	0.134	60.9 ± 14.4	61.7 ± 14.0	0.134
Gender (male)	178 (65.8)	165 (73.0)	0.089	130 (68.1)	135 (70.7)	0.579
Body mass index (kg/m^2^)	23.2 ± 3.4	23.4 ± 3.4	0.505	23.4 ± 3.4	23.3 ± 3.2	0.079
Hypertension	122 (45.2)	112 (49.6)	0.331	84 (44.0)	88 (46.1)	0.681
Diabetes mellitus	83 (30.7)	96 (42.5)	0.007	63 (33.0)	72 (37.7)	0.335
Dyslipidemia	47 (17.4)	56 (24.8)	0.044	38 (19.9)	42 (22.0)	0.615
Current smoking	67 (24.8)	68 (30.1)	0.189	47 (24.6)	60 (31.4)	0.139
Chronic kidney disease	18 (6.7)	20 (8.8)	0.363	15 (7.9)	16 (8.4)	0.851
Previous myocardial infarction	32 (11.9)	30 (13.3)	0.633	23 (12.0)	23 (12.0)	1.000
Previous coronary revascularization	38 (14.1)	40 (17.7)	0.269	29 (15.2)	33 (17.3)	0.579
Previous cerebrovascular accident	25 (9.3)	14 (6.2)	0.207	22 (11.5)	8 (4.2)	0.008
Systolic blood pressure (mmHg)	65.0 ± 27.8	68.0 ± 35.0	0.296	66.4 ± 24.8	65.3 ± 32.5	0.709
Diastolic blood pressure (mmHg)	43.8 ± 23.5	43.8 ± 23.5	0.991	44.4 ± 19.9	42.0 ± 22.0	0.278
Heart rate (beat/min)	83.8 ± 37.2	82.8 ± 39.8	0.771	83.2 ± 35.8	81.0 ± 40.6	0.573
Hemoglobin (mg/dL)	12.3 ± 2.7	12.5 ± 2.8	0.359	12.4 ± 2.6	12.5 ± 2.8	0.595
Total bilirubin (mg/dL)	1.4 ± 3.3	0.8 ± 0.9	0.010	0.9 ± 0.8	0.9 ± 0.9	0.594
Creatinine clearance rate (mL/min/1.73 m^2^)	53.4 ± 35.3	56.1 ± 30.3	0.357	54.6 ± 39.9	57.4 ± 31.2	0.448
Serum glucose (mg/dL)	241.4 ± 131.9	236.2 ± 117.9	0.658	246.4 ± 133.0	233.8 ± 118.8	0.348
NT–proBNP (pg/mL)	10,598.1 ± 11,784.3	8321.0 ± 14,234.5	0.147	11,104.2 ± 12,349.3	8735.5 ± 14,800.7	0.192
Lactic acid (mmol/L)	7.4 ± 3.7	7.6 ± 3.9	0.684	7.4 ± 3.7	7.6 ± 3.9	0.670
Peak CK-MB (ng/mL)	221.3 ± 558.0	212.9 ± 222.4	0.821	243.7 ± 650.9	221.5 ± 227.5	0.657
Cardiopulmonary resuscitation	129 (47.8)	116 (51.3)	0.431	90 (47.1)	105 (55.0)	0.125
Shock to ECMO insertion time (min)	434.7 ± 869.9	358.7 ± 787.0	0.309	389.1 ± 713.8	312.3 ± 673.9	0.280
Initial pump flow (L/min)	3.0 ± 0.8	2.8 ± 0.9	0.132	2.9 ± 0.8	2.8 ± 0.9	0.255
Distal perfusion	97 (35.9)	90 (39.8)	0.372	73 (38.2)	78 (40.8)	0.601
Unloading of left ventricle	55 (20.4)	32 (14.2)	0.070	41 (21.5)	28 (14.7)	0.084
Left ventricular ejection fraction (%)	26.2 ± 12.4	29.0 ± 15.8	0.034	27.0 ± 12.2	27.7 ± 15.0	0.596
Dopamine	161 (59.6)	126 (55.8)	0.384	105 (55.0)	119 (62.3)	0.146
Norepinephrine	170 (63.0)	159 (70.4)	0.083	121 (63.4)	133 (69.6)	0.193
Epinephrine	44 (16.3)	25 (11.1)	0.093	21 (11.0)	24 (12.6)	0.634
Vasopressin	41 (15.2)	21 (9.3)	0.048	28 (14.7)	16 (8.4)	0.054
Vasoactive inotropic score	110.3 ± 176.7	90.4 ± 134.8	0.156	94.3 ± 182.4	93.9 ± 134.1	0.977
Inotropic score	34.1 ± 41.5	17.6 ± 40.9	<0.001	23.6 ± 21.1	20.5 ± 43.8	0.387
Ischemic cardiomyopathy	177 (65.8)	165 (73.0)	0.074	124 (64.9)	141 (73.8)	0.059
Continuous renal replacement therapy	115 (42.6)	80 (35.4)	0.102	85 (44.5)	69 (36.1)	0.095
Mechanical ventilation	226 (83.7)	183 (81.0)	0.426	154 (80.6)	156 (81.7)	0.794

CK-MB = creatine kinase-myocardial band, ECMO = extracorporeal membrane oxygenation, NT–proBNP: N terminal pro B type natriuretic peptide. Values are expressed as n (%) or mean ± standard deviation.

**Table 2 jcm-11-04958-t002:** In-hospital outcomes according to the use of inodilators in the propensity score-matched cohort.

	Overall (n = 382)	Inodilators (n = 191)	No Inodilators (n = 191)	*p* Value
In-hospital cardiac mortality	194 (50.8%)	91 (47.6%)	103 (53.9%)	0.220
In-hospital mortality	168 (44.0%)	79 (41.4%)	89 (46.6%)	0.304
ECMO site bleeding	54 (14.1%)	29 (15.2%)	25 (13.1%)	0.558
Limb ischemia	32 (8.4%)	16 (8.4%)	16 (8.4%)	1.000
Stroke	16 (4.2%)	9 (4.7%)	7 (3.7%)	0.611
GI bleeding	24 (6.3%)	11 (5.8%)	13 (6.8%)	0.674
Sepsis	18 (4.7%)	9 (4.7%)	9 (4.7%)	1.000

ECMO: extracorporeal membrane oxygenation; GI: gastrointestinal.

**Table 3 jcm-11-04958-t003:** Univariate and multivariate analysis of the prediction of in-hospital mortality.

	Univariate Analysis		Multivariate Analysis	
	HR (95% CI)	*p* Value	HR (95% CI)	*p* Value
Use of inodilators	0.768 (0.579–1.018)	0.066	0.702 (0.522–0.944)	0.019
Age (year)	1.024 (1.013–1.034)	<0.001	1.019 (1.005–1.033)	0.007
Gender (male)	0.865 (0.633–1.182)	0.363		
Body mass index (kg/m^2^)	1.035 (0.991–1.080)	0.118		
Systolic blood pressure (mmHg)	0.999 (0.993–1.004)	0.595		
Heart rate	0.995 (0.991–0.999)	0.007	0.998 (0.995–1.002)	0.392
Hypertension	1.504 (1.133–1.996)	0.005	1.196 (0.868–1.646)	0.274
Diabetes mellitus	1.161 (0.871–1.549)	0.309		
Previous chronic kidney disease	1.515 (0.987–2.323)	0.057	0.928 (0.575–1.495)	0.758
Ischemic cardiomyopathy	2.191 (1.546–3.107)	<0.001	1.131 (0.753–1.697)	0.553
Hemoglobin (g/dL)	1.036 (0.981–1.094)	0.204		
Creatinine (mg/dL)	1.083 (0.993–1.180)	0.070		
Creatinine clearance rate (mL/min/1.73 m^2^)	0.994 (0.989–0.999)	0.029	1.001 (0.996–1.006)	0.773
Lactic acid (mmol/L)	1.090 (1.052–1.129)	<0.001	1.077 (1.037–1.119)	<0.001
Peak CK-MB (ng/mL)	1.000 (1.000–1.000)	0.002	1.000 (1.000–1.000)	0.076
Left ventricular ejection fraction (%)	0.984 (0.972–0.995)	0.007	0.986 (0.974–0.999)	0.032
Cardiopulmonary resuscitation	2.229 (1.659–2.995)	<0.001	1.389 (0.993–1.945)	0.055
Shock to ECMO insertion time (min)	1.000 (1.000–1.000)	0.087	1.000 (1.000–1.000)	0.070
Continuous renal replacement therapy	1.910 (1.436–2.540)	<0.001	1.558 (1.138–2.134)	0.006
Mechanical ventilation	6.412 (3.280–12.534)	<0.001	3.266 (1.612–6.616)	0.001
Initial pump flow (L/min)	0.804 (0.663–0.973)	0.025	0.842 (0.689–1.029)	0.094
Distal perfusion	0.644 (0.478–0.866)	0.004	0.692 (0.506–0.946)	0.021
Unloading of left ventricle	1.229 (0.873–1.730)	0.238		
Vasoactive inotropic score	1.001 (1.000–1.001)	0.001	1.001 (1.000–1.002)	0.001
Inotropic score	1.006 (1.004–1.009)	<0.001		
Dopamine	2.033 (1.499–2.756)	<0.001		
Norepinephrine	1.791 (1.292–2.484)	<0.001		
Epinephrine	1.401 (0.949–2.068)	0.090		
Vasopressin	1.531 (1.037–2.261)	0.032		

CI: confidence interval; CK-MB: creatine kinase-myocardial band; ECMO: extracorporeal membrane oxygenation; HR: hazard ratio.

## Data Availability

The data used during the current study are available from the corresponding author on reasonable request.

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
