# Peer review of "Inodilators May Improve the In-Hospital Mortality of Patients with Cardiogenic Shock Undergoing Veno-Arterial Extracorporeal Membrane Oxygenation"

_jcm, 2022, doi:10.3390/jcm11174958_

Round 1
Reviewer 1 Report
Interesting conclusions with the adequate cautions regarding the fact that this study is observational, with some missing details : i.e dosage of inodilators, type of inodilators effectively used.
Author Response
Point 1: Interesting conclusions with the adequate cautions regarding the fact that this study is observational, with some missing details : i.e dosage of inodilators, type of inodilators effectively used.
Response: Thank you for your comment.
Reviewer 2 Report
Manuscript Number: JCM-1834777
Title: The use of inodilators may improve in-hospital mortality inpatients with cardiogenic shock underwent veno-arterial extracorporeal membrane oxygenation.
Author: Mijoo Kim, etc.
1. General Comments
This manuscript, the author investigated inotropes use may improve clinical outcomes in refractory CS patients with VA-ECMO. There is some limitations in this paper, but it is mentioned in the discussion, and there are no additional corrections or remarks from me.
Author Response
Point 1: This manuscript, the author investigated inotropes use may improve clinical outcomes in refractory CS patients with VA-ECMO. There is some limitations in this paper, but it is mentioned in the discussion, and there are no additional corrections or remarks from me.
Response: Thank you for your comment. Actually, we did English editing before the submission. But, we have made extensive English editing again from a native speaker.
Reviewer 3 Report
Dear Sir/Madam,
I had the opportunity to act as a reviewer on the recent submission by Kim et al. to the Journal of Clinical Medicine.
The authors present original research studying the effect of inodilators on clinical outcomes in a mixed patient cohort (registry, observational) with cardiogenic shock undergoing VA-ECMO. They found that use of inodilators was significantly associated with lower in-hospital mortality.
The manuscript is well structured; however, some issues need to be addressed:
- I recommend extensive editing of English language and style, ideally by native speaker.
- Please provide a clear definition of cardiogenic shock. The lines 85-88 read rather as inclusion criteria.
- Regarding tables 1 and 3: what does exactly “unloading of left ventricle” mean – the use of a percutaneous microaxial pump (i.e., Impella)? Please comment.
Best regards,
